# B Cell Lineage in the Human Endometrium: Physiological and Pathological Implications

**DOI:** 10.3390/cells14090648

**Published:** 2025-04-29

**Authors:** Kotaro Kitaya

**Affiliations:** Infertility Center, Iryouhoujin Kouseikai Mihara Hospital, 6-8 Kamikatsura Miyanogo-cho, Nishikyo-ku, Kyoto 615-8227, Japan; kitaya@mihara.com; Tel: +81-75-392-3111; Fax: +81-75-382-3111

**Keywords:** B cells, chronic endometritis, endometriosis, endometrium, infertility, plasma cells

## Abstract

Immunocompetent cells of B lineage function in the humoral immunity system in the adaptive immune responses. B cells differentiate into plasmacytes upon antigen-induced activation and produce different subclasses of immunoglobulins/antibodies. Secreted immunoglobulins not only interact with pathogens to inactivate and neutralize them, but also involve the complement system to exert antibacterial activities and trigger opsonization. Endometrium is a mucosal tissue that lines the mammalian uterus and is indispensable for the establishment of a successful pregnancy. The lymphocytes of B cell lineage are a minority in the human cycling endometrium. Human endometrial B cells have therefore been understudied so far. However, the disorders of the female reproductive tract, including chronic endometritis and endometriosis, have highlighted the importance of further research on the endometrial B cell lineage. This review aims to revisit lymphopoiesis, maturation, commitment, and survival of B cells, shedding light on their physiological and pathological implications in the human endometrium.

## 1. Introduction

B cells (B lymphocytes) are immunocompetent cells that function in the humoral immunity system in the adaptive immune responses. B cells differentiate into plasmacytes (PCs) upon antigen-induced activation and produce different subclasses of immunoglobulins (Ig)/antibodies. Secreted Ig not only interact with pathogens to inactivate and neutralize them, but also involve the complement system to exert antibacterial activities and trigger opsonization (phagocytosis by antibody-coated pathogens) [1].

Endometrium is a mucosal tissue that lines the mammalian uterus and is indispensable for the establishment of a successful pregnancy [2]. Human endometrium contains various types of immunocompetent cells, including T cells, macrophages, natural killer cells, neutrophils, and eosinophils [3]. Endometrial immunocompetent cells are considered not only to provide a front-line defense against invading pathogens in the uterine cavity, but also to play a role in conditioning the local microenvironment for intact pregnancy [4,5,6].

By contrast, lymphocytes of B cell lineage are a minority in the human cycling endometrium [7]. Human endometrial B cells have therefore been understudied so far. However, recent studies on chronic endometritis (CE) and endometriosis have alerted us to the importance of further research on endometrial B cell lineage [8,9]. This review aims to revisit lymphopoiesis, maturation, commitment, and survival of B cells, shedding light on their physiological and pathological roles in the human endometrium.

## 2. B Cell Subpopulations

Based on their ontogeny, phenotype, and functions, B cells are categorized into B-1 cells and B-2 cells [10]. B-1 cells participate in the innate immune system and produce Ig that are distinguished by their recognition of self-antigens and repetitive epitopes such as carbohydrates. B-1 cells are further subdivided into B-1a and B-1b subsets, depending on the presence or absence of the inhibitory surface receptor CD5 [11]. While B-1a cells produce Ig that react with molecular features standard to many microbes and play a pivotal role in scavenging senescent cells, B-1b cells contribute to the acquisition of adaptive immune responses against bacteria without the assistance of T cells. Artificial stimulation of Toll-like receptors on B-1a cells downregulates their CD5 expression, suggesting that B-1b cells are the activated form of B-1a cells [11,12]. B-2 cells are present in the secondary lymphoid organs and are thought to be mediators of adaptive immunity. B-2 cells are subclassified into a predominant population of follicular B (FOB) cells and a minor population of marginal zone B (MZB) cells, both of which are able to undergo Ig class switching and differentiate into memory B cells [13]. While FOB cells eventually differentiate into PCs or long-lived memory B cells, MZB cells reside predominantly within the marginal zone of the spleen, lymph nodes, and peripheral blood. MZB cells rapidly respond to invading antigens from the circulation upon encountering them. MZB cells also exhibit B1 cell-like innate properties and functions. In this manuscript, hereinafter, B2 cells, immature B cells, and memory B cells are referred to as B cells [14].

## 3. B Cell Lymphopoiesis

In mammals, B cells mainly originate in hematopoietic stem cells that reside in the bone marrow and differentiate through a multistep process. Hematopoietic stem cells give rise to B cell progenitors via the stages of lymphomyelo-primed progenitors and common lymphoid progenitors. The maturation of lymphomyelo-primed progenitors into common lymphoid progenitors is induced by Fms-like tyrosine kinase 3 signaling, which upregulates lymphoid-specific genes (E2a, Il7ra, and Ebf1) [15,16] (Figure 1).

Ikaros is a transcription factor in the Ikaros protein family. Ikaros regulates transcriptional programs via the coordination of six zinc finger domains. While the first four zinc finger domains are essential for DNA binding, the last two facilitate homodimerization and heterodimerization with other members of the Ikaros protein family, comprising Aiolos, Helios, Eos, and Pegasus [17]. Ikaros is expressed in the human bone marrow from the early progenitor stage and plays a crucial role in B lymphopoiesis. The lack or deficiency in Ikaros causes a serious reduction in B cells and/or severe hypogammaglobulinemia [18].

B cell commitment is determined by the expression of paired box protein 5 (PAX5), a transcription factor of which gene transcripts first appear at the pro-B cell stage [19]. PAX5 suppresses the expression of other lineage genes, such as notch1 (T cells) or csfr1 (monocytes/macrophages), in common lymphoid progenitors, whereas it upregulates B cell-specific molecules, including CD19, Ig lambda-like polypeptide-5, and CD79A [20]. The expression of PAX5 is maintained until the transitional phase into PCs [21].

Pre-progenitor (pro)-B cells are the earliest stage of B-lineage cells that proceed to the stages of early pro-B cells, and later into precursor (pre)-B cells. Pre-pro-B cells express transcription factors associated with myeloid lineages, including Runt-related transcription factor 2, interferon regulatory factor (IRF)-8, bone marrow stromal antigen 2, and transcription factor 4. These molecules are silenced in the differentiation process into pro-B cells [22].

E2A (also known as transcription factor 3), SRY-related HMG-box 4, and early B-cell factor 1 expressed in pro-B cells drive the expression of recombination activating gene (RAG) 1 and RAG2 [23]. RAG1 and RAG2 form a complex and initiate the T-cell receptor variable–diversity–joining recombination (V(D)J rearrangement) for the production of Ig heavy chains (IgH) [24]. Terminal deoxynucleotidyl transferase plays a role in the diversity acquisition of Ig repertoire by random nucleotide addition to DNA single strands [25]. Another role of early B-cell factor 1 is VpreB1 (CD179A) induction and Ig lambda-like polypeptide-1 (CD179B) production in pro-B cells and early pre-B cells, which are indispensable for the biosynthesis of surrogate light chains [26]. In the maturation process from pro-B cells to pre-B cells, lymphoid enhancer–binding factor 1, a member of the T cell factor/lymphoid enhancer factor family, is involved in cell survival and proliferation via the Wnt signaling pathway. The expression of lymphoid enhancer-binding factor 1 is turned off in mature B cells [27].

Pre-BII large cells are characterized by cytoplasmic IgH, which is paired with surrogate light chains to form pre-B cell receptors (BCRs). In humans, signaling via pre-BCRs, rather than interleukin (IL)-7 receptors, induces a proliferative burst, leading to clonal expansion of pre-BII large cells [16]. Pre-BCR signaling is also a requisite for the differentiation process from pre-BII large cells to pre-BII small cells by regulating positive and negative selection, developmental progression, and allelic exclusion, which is a phenomenon that pre-BII cells use to ensure that one cell does not express two IgH with different specificities [28,29,30]. In the transition into pre-BII small cells, pre-BII large cells rearrange the Ig light chain (IgL) variable–joining genes. Following the successful rearrangement and expression of IgL, previously synthesized IgH and IgL are eventually assembled into BCRs (IgM) [31,32]. The BCR is transported to and anchored on the plasma membrane to form the complex with CD79a (Ig) and CD79b (Ig), which marks the transitional B cells to immature B cells [31,32].

At the development stage of immature B cells, the cell-surface BCR acquires the ability to bind antigens. In the microenvironment of the bone marrow where immature B cells appear, the antigens that engage BCRs are primarily self-antigens. Ligation of BCRs with self-antigens induces signaling responsible for molecular processes to reduce the self-reactivity of immature B cells (central tolerance) [33]. Central tolerance averts the escape of autoreactive immature B cells from the bone marrow. Under these circumstances, residual autoreactive immature B cells are destined for receptor editing (secondary IgL rearrangement), clonal deletion (apoptosis), or anergic state (low BCR expression and signaling) [34]. The regulatory processes of central tolerance, however, are often leaky. More than half of Ig in the immature compartment remain autoreactive or polyreactive [33,34]. Clinical studies estimated that human B lymphopoiesis requires approximately 6–12 months in patients with rheumatoid arthritis using targeted B cell therapies with anti-CD20 monoclonal antibodies (rituximab), B cell reconstitution after hematopoietic stem cell transplantation, or B cell depletion [35,36,37].

## 4. B Cell Maturation

Transitional B cells that migrate from the bone marrow enter into peripheral circulation and eventually differentiate into naïve B cells, FOB cells, or MZB cells. Upon encountering antigens, B cells become proliferative, migrate to the border of the T cell zone or interfollicular region of the secondary lymphoid organs (lymph nodes, tonsils, spleen, or mucosal-associated lymphoid tissues), and are activated [38]. A fraction of activated B cells migrate farther to the extrafollicular foci and differentiate into short-lived plasmablasts that secrete low-affinity antibodies [39]. Some activated B cells proliferate, home to the B-cell follicle, and form germinal centers (GCs). GCs are involved in adaptive immune responses against locally presented antigens. Follicular helper T (Th) cells play an important role in this process. T cell-independent activation promotes the differentiation of FOB and MZB cells to PCs, whereas T cell-dependent activation leads them to memory B cells [40]. In T cell-dependent antigen responses, long-lived CD40L/CD40R interactions between T and B cells differentiate into follicular Th cells and PCs, respectively. The GC reaction typically occurs in the secondary lymphoid organs, but similar structures have also been found in the ectopic or tertiary lymphoid structures [41]. In the GC, B cells undergo somatic hypermutation and subsequent clonal selection. While these B cells increase their Ig affinity against antigens, they differentiate into memory B cells or long-lived PCs (LL-PCs) that return to the bone marrow [1].

The maturation process from B cells into PCs is strictly regulated by a network of multiple transcription factors, including upregulation of PC-specific transcription factors [IRF-4, X-box binding protein-1, and B lymphocyte-induced maturation protein-1] and downregulation of B cell-specific ones [PAX5, IRF8, and BTB Domain and CNC Homolog 2 (BACH2)]. Ensuing events followed by these gene transcriptional alterations are increased cell-surface expression of CD27, CD38, CD138, and CD269, along with concomitant reduction in CD19 and CD20. While transcription factor-associated mechanisms of PC differentiation have been studied intensively, the trajectories that convert naïve B cells into PCs and the regulators that control the fate of PCs remain largely unknown [42].

## 5. B Cell Commitment

Prior to final differentiation into CD138^positive^ nonproliferating PCs, B cells undergo a major change in morphological and epigenomic profiles and become immature and proliferative plasmablasts, the precursor cells of PCs [41]. In the process of the acquisition of Ig secretion potential, the organelles in plasmablasts drastically change. Human circulating plasmablasts are generally defined as CD19^positive^ CD38^bright^ CD27^bright^ cells with high heterogeneity. In vitro study demonstrated that human naïve B cells cultured with IL-2, CD40L, CpG, and anti-IgM Fab’2 proceed to S-phase and display significant gene expression modifications by day 1. These highly proliferative, activated, committed B cells are characterized by the extinction of IL-4/signal transducers and activators of the transcription 6 pathway and Cbl proto-oncogene B ubiquitin ligase expression, downregulation of CD23 on the plasma membrane, along with upregulation of IRF-4. After day 4, under the additional couple of days of maintenance with IL-2, IL-4, and IL-10, a small subset of CD23^negative^ B cells entered the cell cycle after promoting the G1/S transition and differentiated into the plasmablast phenotype [43]. In this process, a striking increase in the expression of the proviral integrations of Moloney virus 2 serine/threonine kinase (PIM2) was identified in these cells, suggesting the role of PIM2 in the B cell commitment [44]. BACH2 is an early regulatory factor that decides the fate and commitment of B cells. Reduction in BACH2 in IgG1 memory B cells accelerates their differentiation into PCs. Following the repression of BACH2, PIM2 expression is stimulated by IL-21, IL-10, IL-6, and signal transducer and activator of the transcription-3 pathway. The high expression of PIM2 at an early stage enables cell cycle entry and G1/S transition of B cells via degradation of cyclin-dependent kinase inhibitor 1B and activation of cell division cycle 25A, promoting the differentiation into plasmablasts. PIM2 also plays a role in protecting cells from apoptosis by suppressing the activation of caspase 3. Finally, an additional 3-day culture with IL-6 and interferon-α led these cells to CD138^positive^ nonproliferating PCs [45].

In mature PCs, the endoplasmic reticulum alters its morphological appearance and peaks in protein production and secretion. These endoplasmic reticulum expansions are mainly led through the inositol-requiring enzyme type-1 pathway by promoting the synthesis of spliced X-box binding protein-1, which, in turn, further activates mitochondrial and endoplasmic reticulum expansions. These unfolded protein responses are also regulated by activating transcription factor 6, an endoplasmic reticulum stress-regulated transmembrane transcription factor, and protein kinase R (PKR)-like endoplasmic reticulum kinase, an endoplasmic reticulum-associated stress sensor protein [43,44,46]. In the course of the final steps of B cell differentiation, mTORC1 plays a pivotal role in the control of the secretory program by PCs, balancing their antibody production and cell survival with the regulation of autophagy to limit endoplasmic reticulum expansions, until amplified and sustained B lymphocyte-induced maturation protein inositol-requiring enzyme type-1/spliced X-box binding protein-1 program takes over. Transcriptomic analysis of B cell commitment to PCs demonstrated that the primary enriched genes in this transition are pro-apoptotic genes associated with cell death (such as *BCL2L11* and *CASP3*) and pro-survival genes related to cell survival (such as *BCL2L1*, *MCL1*, and *XIAP*) [47].

Both memory B cells and LL-PCs are responsible for long-term host immunity against pathogens. While the former cells are recognized as the property that responds to the reinfection of pathogens and their variants, the latter cells are characterized by the protective antibody production [34]. The antigen-driven differentiation from naïve B cells into memory B cells or LL-PCs is induced primally in the B cell follicles and GCs in the secondary lymphoid organs, including lymph nodes, tonsils, spleen, and mucosal-associated lymphoid tissues. First, antigen stimulations via BCRs promote the transition of naïve B cells to GC B cells or short-lived PCs in the B cell follicles. Next, antigen stimulation further drives GC B cells into memory B cells or LL-PCs. Memory B cells are generated in an early immune response, primarily arising from low-affinity cells [48]. Recent studies suggest that the downregulation of Bcl6 and subsequent upregulation of Bcl2 in low-affinity cells may be a key event for their survival [49]. Unlike PC precursors, memory B cell precursors that reside at the periphery of the light zone in the GC are no longer cycling. The transition from GC B cells to memory B cells is thought to include at least three interconnected sequential processes. First, cessation of proliferation in the dark zone is characterized by the stepwise decline in c-Myc expression and active mTORC1. Second, trafficking back to the light zone is accompanied by decay of c-Myc expression and mTORC1 activity, resulting in downregulation of CXC chemokine receptor (CXCR)-4 and migration from the dark zone to the light zone. Third, entrance into the quiescent stage with the acquisition of survival signals is marked by high BACH2 expression required for dampening c-Myc expression and mTORC1 activity [50]. In the following recall responses to antigens, memory B cells differentiate into LL-PCs or re-enter the germinal cell reactions [51].

## 6. B Cell Survival

The number of naïve B cells is tightly controlled under the regulation of signaling via several receptors, including BCRs and B-cell activating factor (BAFF)/BAFF receptor [52]. Distinct from the activating signals induced by antigen binding, the survival signals via the BCR are ligand-independent ones that involve phosphoinositide 3-kinase (PI3K). The key molecule in B cell survival signals is the PI3Kδ isoform, which is expressed in B cells at a higher level than the other three isoforms [53]. Activating signals via BAFF receptor promote protein synthesis and upregulate the expression of some cell cycle-associated proteins, including cyclin D2, cyclin E, and cyclin-dependent kinase 4 in B cells, leading to set them up for potential proliferation in response to mitogenic stimulation [52]. The major mechanism by which BAFF/BAFF receptor manages B cells is mediated through the noncanonical nuclear factor (NF)-B pathway [54]. The BAFF/BAFF receptor combination induces recruitment of tumor necrosis factor receptor-associated factor (TRAF) 3 to the cytoplasmic domain of BAFF receptor, leading to ubiquitination and degradation of TRAF3 by cellular inhibitor of apoptosis protein-1. This results in NF-B inducing kinase-mediated accumulation and phosphorylation of IκB kinase 1, which in turn promotes the phosphorylation of NF-B2, subsequent procession into transcription factor p52, and entry of p52/transcription factor RelB complexes into the nucleus. BAFF receptor-induced ubiquitination and degradation of TRAF3 also stimulate glucose uptake, anaerobic glycolysis, and oxidative phosphorylation in B cells via upregulation of expression of glucose transporter 1 and hexokinase 2, leading to an increase in their cell size. Additionally, the noncanonical NF-B pathway induces the expression of ovarian tumor deubiquitinase 7B to stabilize TRAF3 for a negative feedback loop [54].

Spleen tyrosine kinase (SYK) also plays a critical role in the survival of B cells [55]. SYK is able to bind to phosphorylated immunoreceptor tyrosine-based activation motifs of CD79A and CD79B of BCRs and undergoes BAFF receptor-mediated activation, which induces signaling via extracellular signal-regulated kinase 5 and PI3K/protein kinase B pathways required for B cell survival. Studies using inducible deletion of SYK from mature B cells demonstrated a severe loss of FOB [56]. BCRs and BAFF receptors were suggested to cooperate with CD19 for B cell survival. CD19 is a transmembrane protein expressed in all human B-lineage cells and acts as a coreceptor/adaptor protein for BCRs. CD19 is essential for BAFF receptor-mediated activation of the protein kinase B pathway and following the suppression of proline-directed serine-threonine kinase glycogen synthase kinase (GSK) 3A/3B, and transcription factor forkhead box protein O1. Intriguingly, genetic loss of both GSK3A and GSK3B in mature B cells leads to the loss of both FOB and MZB. In CD19-mediated signaling pathways, mTORC2 also plays a crucial role in B cell maintenance, as its elimination causes a 50% loss of FOB [57].

Wiskott–Aldrich syndrome protein-interacting protein (WIP) is a key molecule in actin cytoskeleton remodeling. WIP not only binds Wiskott–Aldrich syndrome protein, an activator of actin-related protein 2/3 complex, to prevent it from degradation and support its intracellular distribution, but can act independently of Wiskott–Aldrich syndrome protein by promoting actin polymerization and stabilizing actin filaments linking with the adaptor molecules such as noncatalytic region of tyrosine kinase proteins and growth factor receptor-bound protein 2 [58]. WIP also contributes to the intracellular coupling of BAFFR, BCR, and CD19 and is involved in BAFF receptor-mediated phosphorylation of CD19 and activation of the protein kinase B pathway [59].

CD74 is a surface receptor for macrophage migration inhibitory factor. Signaling via CD74 leads to cleavage and release of the CD74 intracellular domain. The nuclear translocation of the CD74 intracellular domain induces gene expression required for B cell survival [60].

GTPase, IMAP family member 1 (GIMAP1 GTPase) is one of the proteins in the GTP-binding superfamily and immuno-associated nucleotide subfamily of nucleotide-binding proteins. GIMAP1 GTPase plays a critical role in the differentiation and development of mature B cells. Loss of GIMAP1 GTPase leads to the complete absence of FOB and MZB in humans. POU domain class 2-associating factor 1 (POU2AF1/BOB1) is a transcriptional coactivator expressed principally in B cells and regulates Ig production and gene expression for CD20. POU2AF1/BOB1 is upregulated in B cells by BAFF/BAFF receptor signaling via noncanonical and canonical NF-B pathways [61].

The attenuation of cellular stress responses associated with the unfolded protein response is essential for the longevity of PCs. Integrated stress response (ISR) is a cellular response equipped in eukaryotic cells to restore homeostasis against diverse internal or environmental stresses such as hypoxia, amino acid/glucose deprivation, and viral infection [62,63]. ISR is drawing attention due to its contribution to cell survival. In ISR, cells activate a common adaptive pathway. Cellular stress is sensed principally by specialized eukaryotic translation initiation factor 2 (EIF2) kinases, including heme-regulated inhibitor, PKR, PKR-like ER kinase (PERK), and general control nonderepressible 2 kinase [64]. Phosphorylation of a single serine by EIF2 kinases induces conversion of EIF2 to the competitive inhibitor of EIF2B complex, a guanine nucleotide exchange factor for EIF2. Inhibition of EIF2B activity by phosphorylated EIF2 lowers cellular EIF2-GTP·Met-tRNAi ternary complex level, leading to the trigger of ISR [65]. B cell differentiation occurs independently of PERK, whereas heme-regulated inhibitor and general control nonderepressible 2 kinase respond to nutrient deprivation and heme deficiency. Of the four EIF2 kinases, PKR is considered a central key regulator for ISR-mediated translation in PCs by utilizing its double-stranded RNA binding domain and PKR-associated activator recruitment [66].

LL-PCs are possibly capable of surviving until the day the host dies without repeated antigen stimulation. According to the results of the somatic hypermutation studies, LL-PCs have been considered to originate in the GCs. However, the identification and analysis of bone marrow LL-PCs suggest that high affinity is not a strict prerequisite for the longevity of PCs [67,68]. While most PCs are distributed in the bone marrow where they can receive survival signals from their surrounding microenvironment, only a few reach survival niches that support their longevity [69,70]. Interestingly, CXCR4 was found to stimulate the extrafollicular maturation of PCs and contribute to the long-term presence of bone marrow LL-PC [71], suggesting a possible role in the interaction between CXCR4 on LL-PCs and CXC ligand 12 (CXCL12) from bone marrow stromal cells in the determination of their fate [72]. PIM2 is essential for the surface expression of CXCR4 in PCs and their migration toward CXCL12 [44]. Human CD138^positive^ PCs, particularly bone marrow PCs, express a high level of cytoplasmic PIM2, suggesting that bone marrow is a preferable microenvironment for PC survival. Thus, PIM2 may play a pivotal role in ISR regulation in LL-PCs. Interestingly, ISR is potentially suppressed by PIM2, inactivating some ISR-associated kinases, including PKR and PERK [44].

## 7. B Cell Lineage in the Human Endometrium

### 7.1. B Cells, Plasmacytes, and Immunoglobulins in Human Nonpathological Endometrium

Human endometrium harbors diverse leucocyte subpopulations including T cells, natural killer cells, macrophages, dendritic cells, and neutrophils. The density and proportion of the endometrial leukocytes fluctuate across the menstrual cycle [2]. Additionally, B cell lineage is a minority in the cycling endometrium in women of reproductive age. B cells account for 2% of whole endometrial cells and only 3% of endometrial lymphocytes [7]. According to their locations, endometrial B cells are classified into two subtypes. First, endometrial single B cells are found scattered throughout the stromal compartment. These endometrial stromal single B cells are resistant to the fluctuations in ovarian steroid levels, as their density is constant throughout the menstrual cycle [73]. Second, some B cells reside in the core of the endometrial lymphoid aggregates, a unique structure in the stratum basalis and deeper part in the stratum functionalis that consists of hundreds of leukocytes, such as inner layer CD4^negative^ CD8^positive^ T cells and outer layer macrophages [74,75]. Contrary to stromal single B cells, lymphoid aggregate–core B cells may be influenced by ovarian steroids, as lymphoid aggregates are more likely to appear in the proliferative phase (before ovulation) than in the secretory phase (after ovulation). Although the significance and function of endometrial lymphoid aggregates remain largely unknown, some studies suggest a role in antigen scavenging and antibody production, similar to GCs in the secondary lymphoid organs. In contrast to other endometrial mononuclear cell subpopulations such as T cells, natural killer cells, and macrophages expressing activated cell markers [high level of CD69 and human leukocyte antigen (HLA)-DR], the majority of endometrial B cells primarily show a rather naïve lymphocyte phenotype with surface expression of IgM and CD22, an inhibitory receptor against BCR signaling, along with a high level of CD19, CD20, CD79A, CD83, CD229, BTG1, and CXCR4 and a moderate level of CD74 and HLA-DR [76]. Flow cytometric analysis discovered CD27^positive^ memory-type B cells in the human endometrium, along with a small number of CD24^negative^ CD38^high^ plasmablast cell types [77].

CD83 is an Ig superfamily glycoprotein expressed in various immunocompetent cells, including B cells. CD83 has one extracellular Ig V-like structure and is anchored on the plasma membrane, and some of them are released as soluble forms. CD83 stabilizes the major histocompatibility complex-class II antigens by negative regulation of the E3 ubiquitin–protein ligases membrane-associated ring-CH-type Finger 1 and 8 [78]. In addition, CD83 was demonstrated to be involved in the longevity of B cells in mice [79]. The expression of CD83 and CD74 may cooperate for the survival of endometrial B cells in the dynamic local microenvironment. Recent studies showed that surface and intracellular expression of CD83 is detectable in endometrial CD19^positive^ B cells, CD11^positive^ dendritic cells, and T cell subpopulations [76]. In vitro stimulation with lipopolysaccharides in cooperation with pregnancy-associated molecules, including estradiol, progesterone, human chorionic gonadotropin, and transforming growth factor-β1 upregulates CD83 expression selectively in endometrial B cells, but not in other endometrial leukocyte subsets and circulating B cells, while dexamethasone reverses it. The hormonal regulation of CD83 expression in endometrial B cells is likely to come from gene expression balancing between this molecule and matrix metalloproteinase 7, supporting the idea that the physiological immunomodulatory role of endometrial B cells in anti-bacterial responses at the fetal–maternal interface during pregnancy [80].

Moreover, endometrial B cells potentially inhibit intrauterine inflammation during gestation and contribute to the prevention of preterm labor. B cells are a minor subpopulation in the decidualized endometrium in early pregnancy, but cells with varying maturation stages are observed in the stromal compartment of the decidua. While all decidualized endometrial stromal B cells express BAFF receptors, decidualized endometrial stromal fibroblasts are capable of producing and secreting BAFF in response to the stimulation by IFN-γ and IFN-α, supporting the idea that the human decidualized endometrial stromal compartment can provide the niche for the survival of local B cells [81]. Indeed, the density of decidualized endometrial stromal B cells slightly increases toward late gestation with some phenotypic and functional changes probably driven by placental hormones [81]. It is noteworthy that the expression level of BAFF in decidualized endometrial stromal fibroblasts is decreased in women with a history of recurrent pregnancy loss compared with those without miscarriage [82]. Interestingly, soluble BAFF receptors are found to be released by decidualized endometrial stromal fibroblasts. The results of in vitro studies suggest that these decidualized endometrial soluble BAFF receptors suppress the proliferation of activated monocytes/macrophages and the secretion of tumor necrosis factor (TNF)-α and IL-6, suggesting their inhibitory role in intrauterine inflammation during gestation [83].

Decidualized endometrial B cells in late pregnancy, particularly at term pregnancy, display higher levels of activated markers of memory B cells than circulating B cells [84]. Recently, mice deficient in B cells were demonstrated to exhibit a lower level of active progesterone-induced blocking factor 1, an anti-inflammatory immunomodulator produced by B cells in the presence of progesterone, in their uteri, and its administration mitigated intrauterine inflammation and reduced preterm labor in these mice. Furthermore, the production of progesterone-induced blocking factor 1 by decidualized endometrial B cells is mediated by IL-33, which in turn suppresses the activation and influx of natural killer cells and neutrophils and the production of proinflammatory cytokines in the decidua [85]. In humans, a decrease in the expression of the IL-33 receptor α chain on decidualized endometrial B cells and a reduction in the production of active progesterone-induced blocking factor 1 in late pregnancy were also associated with preterm labor [85].

Under physiological local conditions, PCs are also minor immunocompetent cells in the stromal compartment in the human cycling endometrium. Approximately 31% of fertile women harbor one or more endometrial stromal plasmacytes (ESPCs) per 10 high power fields under light microscopy [86]. ESPCs are thought to produce Ig subclasses locally. These include heavy chains of IgA_1_, IgA_2_, IgM, IgG_1_, and IgG_2_, along with light chains of Ig, and J chain, which are necessary for the generation of polymeric Igs, whereas IgG_3_, IgG_4_, IgE, and IgD subclasses are not detectable in the nonpathological endometrium [87,88,89]. Two IgA subclasses are expressed constitutively mainly on the apical side of the endometrial surface epithelium, glandular epithelium, and glandular secretion. Similarly, IgM is found on the apical side of the endometrial surface and glandular epithelial areas, but some cells lack its expression, with the variances among individuals. Most IgA- and IgM-bearing endometrial epithelial cells coexpress the J chain and secretory component. These epithelial expression levels of endometrial IgA, IgM, J chain, and secretory components are higher in the proliferative phase compared with the secretory phase. IgG_1_ and IgG_2_ are also localized on the apical side of endometrial epithelial cells, with marked variances within and between individuals. Immunoreactivity to IgM, IgA, IgG_1_, and IgG_2_ in the endometrial stromal compartment with weaker and sparser immunostaining intensity compared with the endometrial epithelial areas [87].

In addition to the local production by ESPC, monomeric IgA (that lacks J chain), IgG_1_, and IgG_2_ are thought to be brought into the endometrial glandular epithelium from the stromal compartment by passive diffusion [89]. Endometrial glandular epithelial cells express HLA-DR across the menstrual cycle regardless of the expression level of the secretory component, implicating the active secretory component-mediated incorporation of polymeric Ig by the endometrial epithelial cells. As shown in other mucosal tissues, these endometrial Ig subclasses display a defensive property against foreign body invasion into the endometrium. Moreover, the menstrual cycle-dependent fluctuation in the expression level of some endometrial Ig subclasses implicates their possible contribution to blastocyst implantation and menstrual mucosal shedding [89].

### 7.2. B Cells, Plasmacytes, and Immunoglobulins in Human Pathological Endometrium

#### 7.2.1. Chronic Endometritis

CE is a localized inflammatory endometrial disorder that is recognized by an unusual infiltration of ESPCs [90]. CE is mostly asymptomatic or oligosymptomatic with some subtle gynecologic symptoms. According to its silent and nondescriptive symptomatologic nature, CE is often missed both by affected women and experienced gynecologists [91]. The principal pathogens of CE include common bacteria in the female urogenital organs: *Mycoplasma* (*M. genitalium* and *M. hominis*), *Ureaplasma* (*U. urealyticum*), *Proteus* species, *Corynebacterium*, *Gardnerella vaginalis*, *Klebsiella pneumoniae*, *Pseudomonas aeruginosa*, *Mycobacterium tuberculosis*, and yeasts (*Saccharomyces cerevisiae* and *Candida*) [92]. Antibiotic treatment against these microorganisms eradicates ESPCs in CE, although multidrug-resistant CE has recently emerged as a clinical problem, like in other medical fields [93,94,95,96,97,98,99] (Table 1).

The increase in densities of ESPCs in the endometrium with CE seems attributable to the selective recruitment of circulating B cells into the endometrium. We investigated the endometrial expression profiles of the chemokines that can induce the selective migratory responses of peripheral blood B cells and PCs (CCR2, CCR7, CXCR1, CXCR2, CXCR4, and CXCR5 ligands) in CE. Of these seven potential chemokines, CXCL13, a CXCR5 ligand that displays a selective chemotactic activity for B cells, is abnormally expressed in endometrial endothelial cells in women with CE, along with CXCL1 in endometrial epithelial cells. In addition, selectin E, a ligand to endoglycan, a sialomucin presented on peripheral blood B cells, is aberrantly expressed in human endometrial endothelial cells in CE [100] (Figure 2). These findings suggest that these local inflammatory microenvironments in CE induce selective extravasation of B cells, rather than PCs, from circulation into the endometrial stromal compartment via endometrial microvessels and attract them further into the epithelial area. These pro-inflammatory molecules are found to be locally induced in isolated endometrial microvascular endothelial cells (CXCL13 and selectin E) and endometrial epithelial cells (CXCL1) in vitro by microbial antigens such as lipopolysaccharide, a Toll-like receptor 2/4 ligand [100]. Moreover, the concentration of IL-6, a differentiation factor of mature B cells, is significantly higher in the menstrual blood of women with CE than in those without CE, indicating that the inflamed endometrium with CE provides extravasated B cells with the niches for in situ differentiation into PCs [101]. Regarding the local Ig profiles, the densities of IgM, IgA_1_, IgA_2_, IgG_1_, and IgG are higher in the stromal compartment during the proliferative phase in the endometrium with CE than those without CE. The endometrial Ig subclasses in CE are characterized by the highest density of IgM^positive^ stromal cells and the predominance of IgG_2_^positive^ stromal cells over IgG_1_^positive^ counterparts [88].

The concentration of TNF-α is also higher in the menstrual effluents of women with CE compared with those without CE [101]. TNF-α is capable of promoting local estrogen biosynthesis in endometrial glandular epithelial cells via estrogen receptor-α signaling, which potentially alters endometrial epithelium to the proliferative phenotype [102] (Figure 3). Studies support the idea that such local microenvironments in CE play a role in the proliferation and survival of the endometrial cell components. Endometrial micropolyposis is a mucosal finding in CE, which is detectable with hysteroscopy [103,104]. The endometrium with micropolyposis has proliferative characteristics and contains a higher concentration of ESPCs than the nonpathologic endometrium. Additionally, one of the histopathological features of CE is the delayed endometrial differentiation in the secretory phase (after ovulation) when the endometrium has to prepare for the implantation of blastocysts. Approximately one-third of the endometrium with CE displays an “out-of-phase” morphological appearance in glandular and surface epithelial cells, i.e., pseudostratification and/or mitotic nuclei, the findings seen in the proliferative phase (before ovulation) [104]. Indeed, the expression levels of messenger RNAs involved in cellular proliferation (*ki-67*), anti-apoptosis (*bcl2* and *bax*), and ovarian steroid receptors (*esr1*, *esr2*, and *pgr*) are abnormally raised in the secretory phase endometrium with CE. By contrast, the expression of messenger RNAs potentially associated with embryo receptivity (*il11*, *ccl4*, *igf1*, and *casp8*) and decidualization (*prl* and *igfbp1*) are reduced, implicating that, in the presence of CE, the endometrium is unable to respond enough to ovarian steroid stimulation and to transform its component cells to acquire the receptivity for implanting blastocysts [105,106,107]. These findings support the idea that the endometrium with CE exhibits progesterone resistance, which is often observed in the endometrium with endometriosis [108].

#### 7.2.2. Endometriosis

One of the unique cellular immunological characteristics seen in the endometrium in endometriosis is the menstrual cycle-independent behavior of endometrial macrophages. Macrophages are subclassified into M1 types (classically activated macrophages) and M2 counterparts (alternatively activated macrophages). M1 macrophages differentiate by the stimulation of interferon-γ and promote inflammatory responses via antigen presentation and production of interleukin IL-6, IL-12, and TNF-α (Figure 3), whereas M2 macrophages differentiate by the stimulation of IL-4 and inhibit inflammatory responses via the production of IL-10, TNF-α, and arginase-I [109]. While whole macrophages increase in density after ovulation in the nonpathological endometrium, such a postovulatory rise of macrophage density is not observed in the eutopic endometrium with endometriosis. However, the density of CD197^positive^ CD80^positive^ M1 macrophages in the eutopic endometrium decreases from stage I to IV, and that of eutopic endometrial CD163^positive^ CD206^positive^ M2 macrophages increases with the progress of the disease [110]. These locally increased eutopic endometrial M2 macrophages are likely to play a role in the proliferation of other endometrial cells and the induction of mucosal angiogenesis. Such biases toward M2 over M1 macrophages are also seen in the ectopic endometrioid lesions of endometriosis [111,112]. The blocking of the actions of IL-6 with antagonistic antibodies can reduce the shift from M1 to M2 counterparts, indicating the putative role of IL-6 in this alteration of local immune responses [111]. In addition, CD16^negative^ CD56^bright^ natural killer cells, which are unusual lymphocyte subpopulations in circulating blood and other organs, increase in number in the endometrium following pituitary luteinizing hormone surge toward the mid/late-secretory-phase endometrium. In the eutopic endometrium with endometriosis, the postovulatory rise of these unique natural killer cells is also seen, but their cytolytic activity is impaired compared with those without endometriosis [113].

Autoimmunity has long been believed to be associated with the onset and progression of endometriosis, but its entity is yet elusive. While multiple studies point out an appearance of the autoantibodies of IgG_1_ and IgG_2_ subclasses against endometrial antigens are detectable in the serum of women with endometriosis, the molecules involved in the autoimmunity in endometriosis remain to be determined [114]. The immunological feature of B cell lineage in the eutopic endometrium of women with endometriosis is the appearance of ESPC and CD5^positive^ CD20^positive^ HLA-DR^positive^ B cells [115], which are commonly found in the endometrium with CE. Although early reports demonstrate that the expression level of IgG in the eutopic endometrium with endometriosis is higher than in those without endometriosis, their subclasses were not detailed. IgG deposits and C3 component of the complement cascade are detectable in the ectopic endometrioid lesions in endometriosis, but their IgG subclasses are not described [116].

BCL6 is a transcriptional repressor that regulates follicular Th cell proliferation and contributes to the development of GC. BCL6 is expressed in the human nonpathological endometrium throughout the menstrual cycle, with a slight increase in the mid-secretory phase. The expression level of BCL6 in the secretory phase was found to be higher in the eutopic endometrium in women with endometriosis and unexplained infertility than in that of fertile women [117]. High BCL6 expression is also reported to be associated with poor reproductive outcomes in in vitro fertilization–embryo transfer cycles in women with unexplained infertility [118]. Additionally, the endometrium in infertile women exposed to more exogenous progesterone had a significantly lower likelihood of high BCL6 expression [119]. BCL6 may therefore be a candidate biomarker for endometriosis and endometrial dysfunction, including progesterone resistance. However, the clinical significance of the medical intervention for infertile women with high BCL6 expression remains controversial. While some studies demonstrated that treatment with leuprorelin, a gonadotropin-releasing hormone agonist, and laparoscopy improved the reproductive outcomes in these women, others found no correlation between BCL6 expression level and live birth rate among normal ovarian responders [120].

Endometriosis lesions are characterized by the infiltration of PC, many of which produce IgM, and macrophages that produce BAFF. Additionally, BAFF concentration was found to be elevated in the serum of endometriosis patients. These findings suggest BAFF-responsive PCs interact with retrograde menstrual tissues to give rise to endometriosis lesions [121]. Intriguingly, a recent report identified mature tertiary lymphoid structures in the ectopic endometrioid lesions on the ovary and fallopian tube in women with endometriosis. These mature tertiary lymphoid structures contained CD3^positive^ CD8^positive^ T cells, CD79a^positive^ B cells, CD208^positive^ dendritic cells, CD21^positive^ follicular dendritic cells, and PNAd^positive^ high endothelial venules, along with immature tertiary lymphoid structures (lacking follicular dendritic cell networks and high endothelial venules) in the eutopic endometrium with or without endometriosis [122].

Furthermore, recent seminal research demonstrated a pathogenic role of *Fusobacterium*, strictly anaerobic Gram-negative rod bacteria, in the development of endometriosis in humans. *Fusobacterium* infiltrate into the human endometrial stroma and activate transforming growth factor-βsignaling, provoking the transition from quiescent stromal fibroblasts to transgelin-expressing myofibroblasts. These myoblasts acquire the ability to proliferate, adhere, and migrate, leading to the formation of the ectopic endometriotic lesions [123]. Antibiotic treatment with metronidazole and/or chloramphenicol prevented the establishment of endometriosis in the *Fusobacterium*-inoculated model mice. Further studies are warranted to elucidate the relationship between B cells and endometriosis [123,124].

#### 7.2.3. Repeated Implantation Failure/Recurrent Pregnancy Loss/Impaired Endometrial Receptivity

Repeated implantation failure (RIF) is an infertile condition where women undergoing IVF-ET treatment fail to have any positive pregnancy tests despite the transfer of at least three good-quality embryos. The major cause of RIF is thought to be chromosomal abnormalities of transferred embryos, but many studies suggest defects in endometrial receptivity, including immunological factors, in this pathology [125]. CE is detectable in 7–34% of infertile women with a history of RIF [126]. A study using single-cell transcriptome profiling demonstrated the selective recruitment of XCR1+ B cells into the endometrium by the action of high levels of various chemokines and endometrial CD49a^positive^ CXCR4^positive^ CCR9^positive^ type 2 natural killer cells in infertile women with a history of RIF [127]. In 2023, the European Society of Human Reproduction and Embryology Working Group issued Good Practice Recommendations on RIF, in which assessment for CE can be considered for infertile women with a history of RIF following IVF-ET, and antibiotic treatment can also be considered if CE is found in these women [128]. Moreover, CE is identifiable in 9–13% of women who experienced unexplained early recurrent (two or more) pregnancy losses. A multicenter, double-blind, randomized clinical trial, termed the CERM (CE and Recurrent Miscarriage) Study, was initiated in the United Kingdom to determine if preconceptual treatment with oral doxycycline administration is effective for increasing live births in women with a history of two or more consecutive first-trimester pregnancy losses [129].

Controlled ovarian stimulation is an indispensable tool in assisted reproductive technology to obtain more mature oocytes available for in vitro fertilization and intracytoplasmic sperm injection. Controlled ovarian stimulation is thought to pose potential detrimental harms on endometrial receptivity, at least in part, due to excessive estradiol exposure and premature progesterone rise [130,131]. Moreover, a prospective cohort study demonstrated that controlled ovarian stimulation brings about not only morphological dyssynchrony and aberrant gene expression profiles, but also unusual endometrial immunocompetent cells. For example, in the periovulatory period, controlled ovarian stimulation induces the infiltration of endometrial monocytes/macrophages, which increase in number during the mid-to-late secretory phase in the natural ovulatory cycle, indicating the “over-advancement” in the local immunologic microenvironment. Additionally, in the window of implantation during the mid-secretory phase, controlled ovarian stimulation provokes uncommon invasion of endometrial B cells and reduction in CD4^positive^ T cells, which is a mucosal cellular immunologic condition similar to CE [132]. Additionally, the proportion of endometrial CD79^positive^ B cells was found to be higher in infertile women with a history of RIF undergoing a failed conception in the subsequent embryo transfer cycles than in those who achieved a successful pregnancy [133].

## 8. Conclusions

A growing body of evidence is accumulating to demonstrate that immunocompetent cells of the B cell lineage are present in the human endometrium under physiological conditions. In several gynecologic diseases that are associated with infertility, the number or density of endometrial B cells/ESPCs is increasing. One of the current questions on endometrial B cell lineage is that we do not have effective ways to distinguish pathologic B cells from physiologic counterparts. Single-cell sequencing analysis prevails in various medical fields. The introduction of these new technologies, in combination with conventional flow cytometry and immunohistochemistry, is awaited to gain our understanding of endometrial B cells in mucosal integrity and pathology. Moreover, little is clear about the functions and roles of endometrial leukocyte aggregates and central B cells. The elucidation of these unique immunologic structures, mainly located in the endometrial basal layer, has the potential to break through the understanding of the mechanisms underlying embryo implantation and fetal–maternal interface formation. Finally, the threshold for ESPC density remains undetermined to define the histopathologic diagnosis of CE.

## Figures and Tables

**Figure 1 cells-14-00648-f001:**
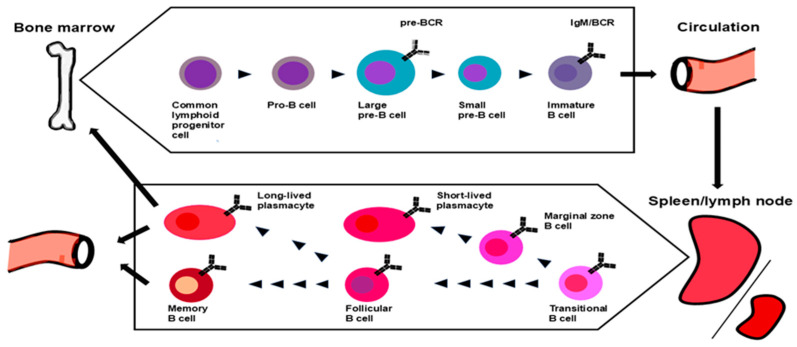
Current models of B cell lymphopoiesis, maturation, and commitment. B cells originate from the bone marrow hematopoietic precursor cells. Recombination-activating gene (RAG) 1/2–induced rearrangement from the germline initiates at the pro-B-cell stage. V-gene segment rearrangement follows in the early pre-B cell stage. Pre-B cell receptor (BCR)–associated signals silence RAG gene expression, suppress the rearrangement of the second H-chains, and induce cell proliferation. IgM-expressing immature B cells alter the gene expression pattern and prepare for egress into the circulation. Immature B cells enter the spleen as transitional B cells, receive survival signals through B-cell activating factor receptor (BAFF-R), and complete the first stage of development, depending on the specificity of their BCR. Upon contact with antigens and support by B helper neutrophils, MZB cells develop into short-lived PC. FOB cells are activated by antigen binding for development in the germinal centers with the aid of helper T cells. T cell-independent activation promotes the differentiation of FOB or MZB cells to PCs, whereas T cell-dependent activation leads them to memory B cells.

**Figure 2 cells-14-00648-f002:**
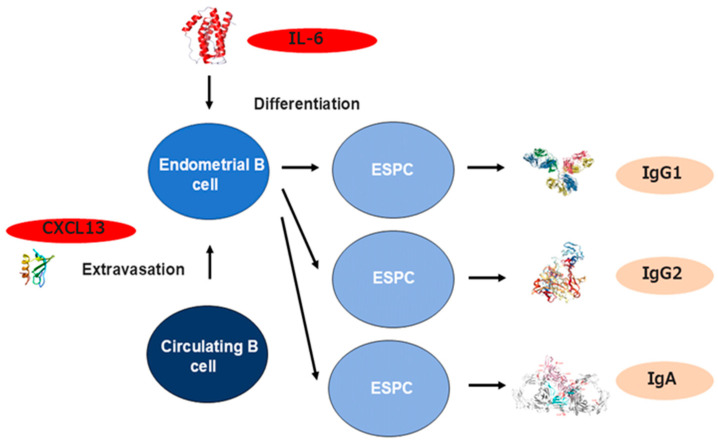
The mechanism underlying B cell recruitment and differentiation in the endometrium with CE. CXCL13 (a CXCR5 antagonist) and CD62E (a ligand for endoglycan) are abnormally expressed in endometrial endothelial cells with CE and display a selective chemotactic activity for B cells. IL-6 (a B cell differentiation factor) concentration is also elevated in the endometrium and induces the differentiation from extravasated B cells to ESPC to produce Ig.

**Figure 3 cells-14-00648-f003:**
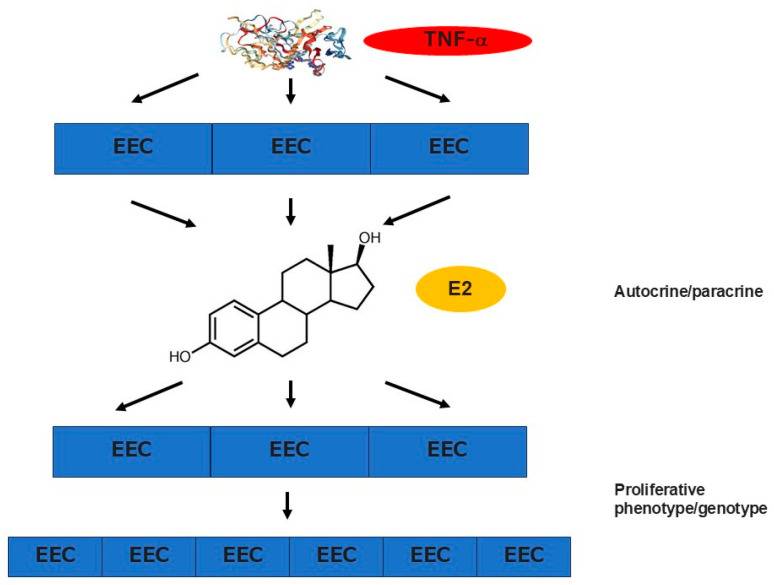
Proliferative phenotype and progesterone resistance in the endometrium with CE and endometriosis. TNF-α level is elevated in the uterine cavity in both pathological conditions. High concentration of TNF-α promotes local estrogen biosynthesis in endometrial glandular epithelial cells via estrogen receptor-α signaling, which in turn potentially alters the epithelial cells to the proliferative phenotype and provokes their survival. For example, the endometrium with CE is characterized by the delayed endometrial differentiation in the secretory phase (after ovulation) when this mucosal tissue has to prepare for the implantation of blastocysts. In addition, approximately one-third of the endometrium with CE displays an “out-of-phase” morphological appearance in glandular and surface epithelial cells, i.e., pseudostratification and/or mitotic nuclei, the findings seen in the proliferative phase (before ovulation). While the endometrial expression levels of messenger RNAs involved in cellular proliferation (*ki-67*), anti-apoptosis (*bcl2* and *bax*), and ovarian steroid receptors (*esr1*, *esr2*, and *pgr*) are abnormally raised in the secretory phase, the expression of messenger RNAs potentially associated with embryo receptivity (*il11*, *ccl4*, *igf1*, and casp8) and decidualization (*prl* and *igfbp1*) are reduced, supporting the idea that the endometrium is unable to respond enough to ovarian steroid stimulation and to transform its component cells to acquire the receptivity for implanting blastocysts.

**Table 1 cells-14-00648-t001:** Inflammatory and immunologic signature in CE.

-Infiltration of CD138^positive^ ESPC, along with CD20^positive^ B cells in the endometrial surface/glandular epithelium and gland secretion
-Upregulation of the molecules associated with B cell extravasation (CXCL13 and CD62E) in endometrial microvascular endothelium and B cell chemotaxis (CXCL1) in the endometrial stroma.
-Local accumulation of IgG_1_, IgG_2_, IgM, IgA_1_, and IgA_2_ with a predominance of IgG_2_
-Elevated concentration of proinflammatory cytokines, including TNF-α, IL-1, and IL-6, in the uterine cavity
-Lack of systemic inflammatory responses, including fever, leukocytosis, and C-reactive protein rise.
-Unlikely association with IgG_4_-related disease.

## Data Availability

No new data were created or analyzed in this study.

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
