# Peer review of "B Cell Lineage in the Human Endometrium: Physiological and Pathological Implications"

_cells, 2025, doi:10.3390/cells14090648_

Round 1
Reviewer 1 Report
Comments and Suggestions for Authors
Main Question:
The manuscript addresses the role of B-lymphocytes in the physiology and pathogenesis of the reproductive system.
Originality and Relevance:
The topic is highly relevant to the field. The role of B-lymphocytes in the physiology and pathogenesis of the reproductive system has not been explored in such detail in any previous review. Therefore, the manuscript fills a specific gap in the current scientific literature.
Contribution to the Field:
The manuscript provides a detailed description of the role of B-lymphocytes, particularly focusing on their mechanisms of action during inflammation in the reproductive system. This adds significant value compared to previously published material.
Methodological Improvements:
I have no methodological comments or suggestions for improvement.
Consistency of Conclusions:
The conclusions are consistent with the evidence and arguments presented throughout the manuscript. They effectively summarize the key biological issues and offer direction for future research.
References:
Yes, the references are appropriate and relevant to the subject matter.
Additional Comments:
The only one suggestion is to explain the abbreviations used in the figures.
Author Response
Comment. Main Question: The manuscript addresses the role of B-lymphocytes in the physiology and pathogenesis of the reproductive system. Originality and Relevance: The topic is highly relevant to the field. The role of B-lymphocytes in the physiology and pathogenesis of the reproductive system has not been explored in such detail in any previous review. Therefore, the manuscript fills a specific gap in the current scientific literature. Contribution to the Field: The manuscript provides a detailed description of the role of B-lymphocytes, particularly focusing on their mechanisms of action during inflammation in the reproductive system. This adds significant value compared to previously published material. Methodological Improvements: I have no methodological comments or suggestions for improvement. Consistency of Conclusions: The conclusions are consistent with the evidence and arguments presented throughout the manuscript. They effectively summarize the key biological issues and offer direction for future research. References: Yes, the references are appropriate and relevant to the subject matter.
Answer I thank the reviewer for all his/her efforts to read the manuscript thoroughly in the limited time and provide all these encouraging comments. Endometrial NK cells, T cell subsets, macrophages, but not B cells, have been extensively researched by lots of experts. I hope this manuscript will help potential future researchers to look at the B cell lineage in this mucosa if it is published.
Additional Comments: The only one suggestion is to explain the abbreviations used in the figures.
In Figure 1 legend, the abbreviations that appear for the first time in the manuscript were spelled out. These include RAG (recombination-activating gene), BCR (B cell receptor), and BAFF-R (B-cell activating factor receptor). (Line 74-84).
Reviewer 2 Report
Comments and Suggestions for Authors
This review is appreciated and addresses and important and largely unexplored topic.
I have some concerns and suggestions:
Major:
- The first paragraphs (1-6) until line 413 should be significantly shorted. The author should rather refer to some excellent reviews regarding B cell development, maturation etc. This section should be finally focused on the B cell types that are actually found in the endometrium. Morover, the focus should be on human B cells.
- Paragraph 7 should include a rather detailed figure of the anatomy and microanatomy of the human endometrium under the different conditions (healthy, term/non-term, pregnancy, inglamed) and include the immune cell population found therein, and a particular description of the B cells. it woill be important to include the structures of the different vessels (lymphatics, blood) and lymphoid structures. Here, I suggest that data from mice and human may be mixed, however, this should be delineated clearly.
- Paragraph 2, B cell subpopulations: MZB cells are missing. I do not think that regulatory B cells are subpopulations; they represent transient cell types that arise during development or activation. Is the limited repertoire of B1 cells really a consequence of impairedf affinity maturation and class switching? Class switching does not influence the repertoire. We talk here about the primary repertoire. Its limitation is probably a consequence of B1 cell development. Are MZB cells really characterized by their longevity and do they arise early in life? Please provide references for this statement.
- Paragraph 4: Its is unclear why suddenly Germinal center dynamics adn CLL are introduced (l 229-246). I find this pararaph rather confusing.
- Paragraph 7, lines 450-459: human or mouse data? L 463, varyingff maturation stages: which ones? L. 506: ...monomeric IgA that lacks the expression of Jchain: antibodies cannot express anything; the author means that monomeric IgA lacks Jchain?
- L. 637 ff: I do not think that Bcl6 expression data are relevant unless they refer to the Bcl6 protein (because Bcl6 is regulated on the protein level). Please clarify.
Minor:
- The frequent and appropriate use of the word "meanwhile" should be reconsidered.
- What does the word "Decidual" mena? Does the author mean resident?
Author Response
Reviewer 2
Overall comment. This review is appreciated and addresses and important and largely unexplored topic.
Answer to overall comment. I thank the reviewer for all the efforts to read the manuscript thoroughly in the limited time and provide all these helpful comments for revisions.
Major
Comment 1. The first paragraphs (1-6) until line 413 should be significantly shorted. The author should rather refer to some excellent reviews regarding B cell development, maturation etc. This section should be finally focused on the B cell types that are actually found in the endometrium. Morover, the focus should be on human B cells.
Answer 1. The sections 1-6 were reduced from 413 lines to 326 lines by focusing on the topics potentially associated with “human endometrial” B cells.” The references were deleted and added (reference No.10) accordingly. (Line 46-49 and 55-60).
Comment 2. Paragraph 7 should include a rather detailed figure of the anatomy and microanatomy of the human endometrium under the different conditions (healthy, term/non-term, pregnancy, inglamed) and include the immune cell population found therein, and a particular description of the B cells. it woill be important to include the structures of the different vessels (lymphatics, blood) and lymphoid structures. Here, I suggest that data from mice and human may be mixed, however, this should be delineated clearly.
Answer 2. Thank you for these reasonable comments. The section 7 was revised with a focus on human studies. The animal studies are included only when they help understand “human” endometrial B cells. (Line 46-49 and 55-60) Please understand that I gave up the descriptions on endometrial anatomy due to the limited 10-day revision period.
Comment 3. Paragraph 2, B cell subpopulations: MZB cells are missing. I do not think that regulatory B cells are subpopulations; they represent transient cell types that arise during development or activation. Is the limited repertoire of B1 cells really a consequence of impairedf affinity maturation and class switching? Class switching does not influence the repertoire. We talk here about the primary repertoire. Its limitation is probably a consequence of B1 cell development. Are MZB cells really characterized by their longevity and do they arise early in life? Please provide references for this statement.
Answer 3. I am so grateful for these comments. As pointed out here, the article that classified regulatory B cells as a B cell subpopulation is a minority. In addition, the first version included some misunderstandings on class switching and MZB cells. They were deleted from the manuscript. (Line 46-49 and 55-60).
Comment 4. Paragraph 4: Its is unclear why suddenly Germinal center dynamics adn CLL are introduced (l 229-246). I find this pararaph rather confusing.
Answer 4. The review comment on the sudden confusing appearance of the topics on GC dynamics and CLL is quite right. These parts were deleted. (Line 147-173)
Comment 5. Paragraph 7, lines 450-459: human or mouse data? L 463, varyingff maturation stages: which ones? L. 506: ...monomeric IgA that lacks the expression of Jchain: antibodies cannot express anything; the author means that monomeric IgA lacks Jchain? 637 ff: I do not think that Bcl6 expression data are relevant unless they refer to the Bcl6 protein (because Bcl6 is regulated on the protein level). Please clarify.
Answer 5. Thank you for these comments. About Line 450-459, this is our study on humans. This was clarified. (Line 455). About Line 463, due to the limited sample volumes, we failed to identify the maturation stages of endometrial B cells/PCs. The text was therefore rewritten. (Line 465-466)
Comment 6. About Line 506. The words were corrected as “monomeric IgA (that lacks J chain)”. (Line 422)
Answer 6. About Line 637. I limited the quoted articles (ref 117-120) to the protein expression level. The papers that are irrelevant to BCL6 protein were excluded. (Line 553-568)
Minor:
Comment 1. The frequent and appropriate use of the word "meanwhile" should be reconsidered.
Answer 1. Thank you for your comments on grammatical correction. Unnecessary or out-of-place uses of ‘meanwhile” were deleted or replaced with other words. In the revised version, “meanwhile” appears only twice. (Line 258 and 332)
Comment 2. What does the word "Decidual" mena? Does the author mean resident?
Answer 2. The term “decidua” is frequently used in the field of obstetrics and genecology and refers to “modification of uterine mucosal lining of the uterus that is in preparation for pregnancy. For clarification, the term “decidualized endometrium” was adopted throughout the menstrual cycle (Line 377-402).